# Exemplar VAE: Linking Generative Models, Nearest Neighbor Retrieval, and Data Augmentation

**Sajad Norouzi**[1,2]  **David J. Fleet**[1,2,3]  **Mohamamd Norouzi**[3]
sajadn@cs.toronto.edu  fleet@cs.toronto.edu  mnorouzi@google.com
[1]University of Toronto,  [2]Vector Institute,  [3]Google Research

## Abstract

We introduce Exemplar VAEs, a family of generative models that bridge the gap between *parametric* and *non-parametric, exemplar based* generative models. Exemplar VAE is a variant of VAE with a non-parametric prior in the latent space based on a Parzen window estimator. To sample from it, one first draws a random exemplar from a training set, then stochastically transforms that exemplar into a latent code and a new observation. We propose retrieval augmented training (RAT) as a way to speed up Exemplar VAE training by using approximate nearest neighbor search in the latent space to define a lower bound on log marginal likelihood. To enhance generalization, model parameters are learned using exemplar leave-one-out and subsampling. Experiments demonstrate the effectiveness of Exemplar VAEs on density estimation and representation learning. Importantly, generative data augmentation using Exemplar VAEs on permutation invariant MNIST and Fashion MNIST reduces classification error from 1.17% to 0.69% and from 8.56% to 8.16%. Code is available at https://github.com/sajadn/Exemplar-VAE.

## 1 Introduction

*Non-parametric, exemplar based* methods use large, diverse sets of exemplars, and relatively simple learning algorithms such as Parzen window estimation [44] and CRFs [34], to deliver impressive results on image generation (*e.g.,* texture synthesis [15], image super resolution [16], and inpaiting [10, 25]). These approaches generate new images by randomly selecting an exemplar from an existing dataset, and modifying it to form a new observation. Sample quality of such models improves as dataset size increases, and additional training data can be incorporated easily without further optimization. However, exemplar based methods require a distance metric to define neighborhood structures, and metric learning in high dimensional spaces is a challenge in itself [28, 57].

Conversely, conventional *parametric* generative models based on deep neural nets enable learning complex distributions (*e.g.,* [43, 47]). One can use standard generative frameworks [13, 14, 18, 32, 49] to optimize a decoder network to convert noise samples drawn from a factored Gaussian distribution into real images. When training is complete, one would discard the training dataset and generate new samples using the decoder network alone. Hence, the burden of generative modeling rests entirely on the model parameters, and additional data cannot be incorporated without training.

This paper combines the advantages of exemplar based and parametric methods using amortized variational inference, yielding a new generative model called Exemplar VAE. It can be viewed as a variant of Variational Autoencoder (VAE) [32, 49] with a non-parametric Gaussian mixture (Parzen window) prior on latent codes.

To sample from the Exemplar VAE, one first draws a random exemplar from a training set, then stochastically transforms it into a latent code. A decoder than transforms the latent code into a new observation. Replacing the conventional Gaussian prior into a non-parameteric Parzen window improves the representation quality of VAEs as measured by kNN classification, presumably because a Gaussian mixture prior with many components captures the manifold of images and their attributes better. Exemplar VAE also improves density estimation on MNIST, Fashion MNIST, Omniglot, and CelebA, while enabling controlled generation of images guided by exemplars.

We are inspired by recent work on generative models augmented with external memory (*e.g.,* [23, 37, 55, 30, 4]), but unlike most existing work, we do not rely on pre-specified distance metrics to define neighborhood structures. Instead, we simultaneously learn an autoencoder, a latent space, and a distance metric by maximizing log-likelihood lower bounds. We make critical technical contributions to make Exemplar VAEs scalable to large datasets, and enhance their generalization.

The main contributions of this paper are summarized as follows:
1. We introduce Exemplar VAE along with critical regularizers that combat overfitting;
2. We propose *retrieval augmented training (RAT)*, using approximate nearest neighbor search in the latent space, to speed up training based on a novel log-likelihood lower bound;
3. Experimental results demonstrate that Exemplar VAEs consistently outperform VAEs with a Guassian prior or VampPrior [55] on density estimation and representation learning;
4. We demonstrate the effectiveness of generative data augmentation with Exemplar VAEs for supervised learning, reducing classification error of permutation invariant MNIST and Fashion MNIST significantly, from $1.17\%$ to $0.69\%$ and from $8.56\%$ to $8.16\%$ respectively.

## 2   Exemplar based Generative Models

By way of background, an exemplar based generative model is defined in terms of a dataset of $N$ exemplars, $X \equiv \{\mathbf{x}_n\}_{n=1}^N$, and a parametric transition distribution, $T_\theta(\mathbf{x} \mid \mathbf{x}')$, which stochastically transforms an exemplar $\mathbf{x}'$ into a new observation $\mathbf{x}$. The log density of a data point $\mathbf{x}$ under an exemplar based generative model $\{X, T_\theta\}$ can be expressed as

$$\log p(\mathbf{x} \mid X, \theta) \;=\; \log \sum_{n=1}^N \frac{1}{N} T_\theta(\mathbf{x} \mid \mathbf{x}_n) \,, \tag{1}$$

where we assume the prior probability of selecting each exemplar is uniform. Suitable transition distributions should place considerable probability mass on the reconstruction of an exemplar from itself, *i.e.,* $T_\theta(\mathbf{x} \mid \mathbf{x})$ should be large for all $\mathbf{x}$. Further, an ideal transition distribution should be able to model the conditional dependencies between different dimensions of $\mathbf{x}$ given $\mathbf{x}'$, since the dependence of $\mathbf{x}$ on $\mathbf{x}'$ is often insufficient to make dimensions of $\mathbf{x}$ conditionally independent.

One can view the Parzen window or Kernel Density estimator [44], as a simple type of exemplar based generative model in which the transition distribution is defined in terms of a prespecified kernel function and its meta-parameters. With a Gaussian kernel, a Parzen window estimator takes the form

$$\log p(\mathbf{x} \mid X, \sigma^2) \;=\; -\log C - \log N + \log \sum_{n=1}^N \exp \frac{-\|\mathbf{x} - \mathbf{x}_n\|^2}{2\sigma^2} \,, \tag{2}$$

where $\log C = d_x \log(\sqrt{2\pi}\sigma)$ is the log normalizing constant of an isotropic Gaussian in $d_x$ dimensions. The non-parametric nature of Parzen window estimators enables one to exploit extremely large heterogeneous datasets of exemplars for density estimation. That said, simple Parzen window estimation typically underperforms parametric density estimation, especially in high dimensional spaces, due to the inflexibility of typical transition distributions, *e.g.,* when $T(\mathbf{x} \mid \mathbf{x}') = \mathcal{N}(\mathbf{x} \mid \mathbf{x}', \sigma^2 I)$.

This work aims to adopt desirable properties of non-parametric exemplar based models to help scale parametric models to large heterogeneous datasets and representation learning. In effect, we learn a latent representation of the data for which a Parzen window estimator is an effective prior.

## 3   Exemplar Variational Autoencoders

The generative process of an Exemplar VAE is summarized in three steps:
1. Sample $n \sim \mathrm{Uniform}(1, N)$ to obtain a random exemplar $\mathbf{x}_n$ from the training set, $X \equiv \{\mathbf{x}_n\}_{n=1}^N$.
2. Sample $\mathbf{z} \sim r_\phi(\cdot \mid \mathbf{x}_n)$ using an exemplar based prior, $r_\phi$, to transform an exemplar $\mathbf{x}_n$ into a distribution over latent codes, from which $\mathbf{z}$ is drawn.
3. Sample $\mathbf{x} \sim p_\theta(\cdot \mid \mathbf{z})$ using a decoder to transform $\mathbf{z}$ into a distribution over observations, from which $\mathbf{x}$ is drawn.

Accordingly, the Exemplar VAE can be interpreted as a variant of exemplar based generative models in (1) with a parametric transition function defined in terms of a latent variable $\mathbf{z}$, *i.e.,*

$$T_{\phi,\theta}(\mathbf{x} \mid \mathbf{x}') \;=\; \int_z r_\phi(\mathbf{z} \mid \mathbf{x}') \, p_\theta(\mathbf{x} \mid \mathbf{z}) \, d\mathbf{z} \,. \tag{3}$$

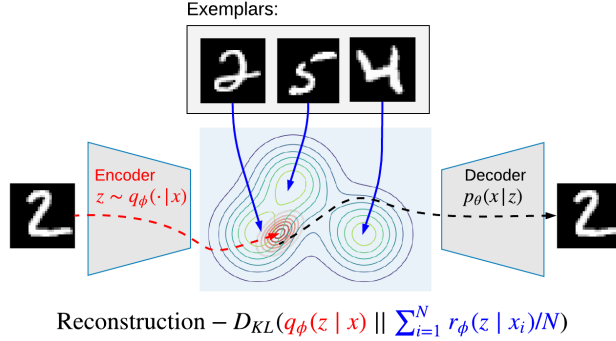

Exemplars:

Reconstruction $- D_{KL}(q_\phi(z \mid x) \parallel \sum_{i=1}^{N} r_\phi(z \mid x_i)/N)$

Figure 1: Exemplar VAE is a type of VAE with a non-parametric mixture prior in the latent space. Here, only 3 exemplars are shown, but the set of exemplars often includes thousands of data points from the training dataset. The objective function is similar to a standard VAE with the exception that the KL term measures the disparity between the variational posterior $q_\phi(\mathbf{z} \mid \mathbf{x})$ and a mixture of exemplar based priors $\sum_{n=1}^{N} r_\phi(\mathbf{z} \mid \mathbf{x}_n)/N$.

This model assumes that, conditioned on $\mathbf{z}$, an observation $\mathbf{x}$ is independent from an exemplar $\mathbf{x}'$. This conditional independence simplifies the formulation, enables efficient optimization, and encourages a useful latent representation.

By marginalizing over the exemplar index $n$ and the latent variable $\mathbf{z}$, one can derive an evidence lower bound (ELBO) [3, 29] on log marginal likelihood for a data point $\mathbf{x}$ as follows (derivation in section F of supplementary materials):

$$
\log p(\mathbf{x}; X, \theta, \phi) = \log \sum_{n=1}^{N} \frac{1}{N} T_{\phi,\theta}(\mathbf{x} \mid \mathbf{x}_n) = \log \sum_{n=1}^{N} \frac{1}{N} \int_z r_\phi(\mathbf{z} \mid \mathbf{x}_n) \, p_\theta(\mathbf{x} \mid \mathbf{z}) \, d\mathbf{z} \quad (4)
$$

$$
\geq \underbrace{\mathbb{E}_{q_\phi(\mathbf{z} \mid \mathbf{x})} \log p_\theta(\mathbf{x} \mid \mathbf{z})}_{\text{reconstruction}} - \underbrace{\mathbb{E}_{q_\phi(\mathbf{z} \mid \mathbf{x})} \log \frac{q_\phi(\mathbf{z} \mid \mathbf{x})}{\sum_{n=1}^{N} r_\phi(\mathbf{z} \mid \mathbf{x}_n)/N}}_{\text{KL term}}
$$

$$
= O(\theta, \phi; \mathbf{x}, X). \quad (5)
$$

We use (5) as the Exemplar VAE objective to optimize parameters $\theta$ and $\phi$. Note that $O(\theta, \phi; \mathbf{x}, X)$ is similar to the ELBO for a standard VAE, the difference being the definition of the prior $p(\mathbf{z})$ in the KL term. The impact of exemplars on the learning objective can be summarized in the form of a mixture model prior in the latent space, with one mixture component per exemplar, *i.e.,* $p(\mathbf{z} \mid X) = \sum_n r_\phi(\mathbf{z} \mid \mathbf{x}_n)/N$. Fig. 1 illustrates the training procedure and objective function for Exemplar VAE.

A VAE with a Gaussian prior uses an encoder during training to define a variational bound [32]. Once training is finished, new observations are generated using the decoder network alone. To sample from an Exemplar VAE, we need the decoder and access to a set of exemplars and the exemplar based prior $r_\phi$. Importantly, given the non-parametric nature of Exemplar VAEs, one can train this model with one set of exemplars and perform generation with another, potentially much larger set.

As depicted in Figure 1, the Exemplar VAE employs two encoder networks, *i.e.,* $q_\phi(\mathbf{z} \mid \mathbf{x})$ as the variational posterior, and $r_\phi(\mathbf{z} \mid \mathbf{x}_n)$ for mapping an exemplar $\mathbf{x}_n$ to the latent space for the exemplar based prior. We adopt Gaussian distributions for both $q_\phi$ and $r_\phi$. To ensure that $T(\mathbf{x} \mid \mathbf{x})$ is large, we share the means of $q_\phi$ and $r_\phi$. This is also inspired by the VampPrior [55] and discussions of the aggregated variational posterior as a prior [40, 27]. Accordingly, we define

$$
q_\phi(\mathbf{z} \mid \mathbf{x}) = \mathcal{N}(\mathbf{z} \mid \boldsymbol{\mu}_\phi(\mathbf{x}), \, \Lambda_\phi(\mathbf{x})), \quad (6)
$$

$$
r_\phi(\mathbf{z} \mid \mathbf{x}_n) = \mathcal{N}(\mathbf{z} \mid \boldsymbol{\mu}_\phi(\mathbf{x}_n), \, \sigma^2 I). \quad (7)
$$

The two encoders use the same parametric mean function $\boldsymbol{\mu}_\phi$, but they differ in their covariance functions. The variational posterior uses a data dependent diagonal covariance matrix $\Lambda_\phi$, while the exemplar based prior uses an isotropic Gaussian (per exemplar), with a shared, scalar parameter $\sigma^2$. Accordingly, $\log p(\mathbf{z} \mid X)$, the log of the aggregated exemplar based prior is given by

$$
\log p(\mathbf{z} \mid X) = -\log C' - \log N + \log \sum_{j=1}^{N} \exp \frac{-\|\mathbf{z} - \boldsymbol{\mu}_\phi(\mathbf{x}_j)\|^2}{2\sigma^2}, \quad (8)
$$

where $\log C' = d_z \log(\sqrt{2\pi}\sigma)$. Recall the definition of Parzen window estimator with a Gaussian kernel in (2), and note the similarity between (2) and (8). The Exemplar VAE's Gaussian mixture prior is a Parzen window estimate in the latent space, hence the Exemplar VAE can be interpreted as a *deep* variant of Parzen window estimation.

The primary reason to adopt a shared $\sigma^2$ across exemplars in (7) is computational efficiency. Having a shared $\sigma^2$ enables parallel computation of all pairwise distances between a minibatch of latent codes $\{\mathbf{z}_b\}_{b=1}^B$ and Gaussian means $\{\boldsymbol{\mu}_\phi(\mathbf{x}_j)\}_{j=1}^N$ using a single matrix product. It also enables the use of existing approximate nearest neighbor search methods for Euclidean distance (*e.g.,* [42]) to speed up Exemplar VAE training, as described next.

## 3.1 Retrieval Augmented Training (RAT) for Efficient Optimization

The computational cost of training an Exemplar VAE can become a burden as the number of exemplars increases. This can be mitigated with fast, approximate nearest neighbor search in the latent space to find a subset of exemplars that exert the maximum influence on the generation of each data point. Interesting, as shown below, the use of approximate nearest neighbor for training Exemplar VAEs is mathematically justified based on a lower bound on the log marginal likelihood.

The most costly step in training an Exemplar VAE is in the computation of $\log p\left(\mathbf{z} \mid X\right)$ in (8) given a large dataset of exemplars $X$, where $\mathbf{z} \sim q_\phi(\mathbf{z} \mid \mathbf{x})$ is drawn from the variational posterior of $\mathbf{x}$. The rest of the computation, to estimate the reconstruction error and the entropy of the variational posterior, is the same as a standard VAE. To speed up the computation of $\log p\left(\mathbf{z} \mid X\right)$, we evaluate $\mathbf{z}$ against $K \ll N$ exemplars that exert the maximal influence on $\mathbf{z}$, and ignore the rest. This is a reasonable approximation in high dimensional spaces where only the nearest Gaussian means matter in a Gaussian mixture model. Let $\mathrm{kNN}(\mathbf{z}) \equiv \{\pi_k\}_{k=1}^K$ denote the set of $K$ exemplar indices with approximately largest $r_\phi(\mathbf{z} \mid \mathbf{x}_{\pi_k})$, or equivalently, the smallest $\|\mathbf{z} - \boldsymbol{\mu}_\phi(\mathbf{x}_{\pi_k})\|^2$ for the model in (7). Since probability densities are non-negative and $\log$ is monotonically increasing, it follows that

$$\log p\left(\mathbf{z} \mid X\right) \;=\; -\log N + \log \sum_{j=1}^N r_\phi(\mathbf{z} \mid \mathbf{x}_j) \;\geq\; -\log N + \log \sum_{k \in \mathrm{kNN}(\mathbf{z})} r_\phi(\mathbf{z} \mid \mathbf{x}_{\pi_k}) \qquad (9)$$

As such, approximating the exemplar prior with approximate kNN is a lower bound on (8) and (5).

To avoid re-calculating $\{\boldsymbol{\mu}_\phi(\mathbf{x}_j)\}_{j=1}^N$ for each gradient update, we store a cache table of most recent latent means for each exemplar. Such cached latent means are used for approximate nearest neighbor search to find $\mathrm{kNN}(\mathbf{z})$. Once approximate kNN indices are found, the latent means, $\{\boldsymbol{\mu}_\phi(\mathbf{x}_{\pi_k})\}_{k \in \mathrm{kNN}(\mathbf{z})}$, are re-calculated to ensure that the bound in (9) is valid. The cache is updated whenever a new latent mean of a training point is available, *i.e.,* we update the cache table for any point covered by the training minibatch or the kNN exemplar sets. Section C in the supplementary materials summaries the Retrieval Augmented Training (RAT) procedure.

## 3.2 Regularizing the Exemplar based Prior

Training an Exemplar VAE by simply maximizing $O(\theta, \phi; \mathbf{x}, X)$ in (5), averaged over training data points $\mathbf{x}$, often yields massive overfitting. This is not surprising, since a flexible transition distribution can put all its probability mass on the reconstruction of each exemplar, *i.e.,* $p(\mathbf{x} \mid \mathbf{x})$, yielding high log-likelihood on training data but poor generalization. Prior work [4, 55] also observed such overfitting, but no remedies have been provided. To mitigate overfitting we propose two simple but effective *regularization* strategies:

1. **Leave-one-out during training.** The generation of a given data point is expressed in terms of dependence on all exemplars except that point itself. The non-parametric nature of the generative model enables easy adoption of such a leave-one-out (LOO) objective during training, to optimize

$$O_1(\phi, \theta; X) \;=\; \sum_{i=1}^N \log \sum_{n=1}^N \frac{\mathbb{1}_{[i \neq n]}}{N-1} T_{\phi,\theta}(\mathbf{x}_i \mid \mathbf{x}_n)\,, \qquad (10)$$

   where $\mathbb{1}_{[i \neq n]} \in \{0, 1\}$ is an indicator function, taking the value of 1 if and only if $i \neq n$.

2. **Exemplar subsampling.** Beyond LOO, we observe that explaining a training point using a subset of the remaining training exemplars improves generalization. To that end, we use a hyper-parameter $M$ to define the exemplar subset size for the generative model. To generate $\mathbf{x}_i$ we draw $M$ indices

$\pi \equiv \{\pi_m\}_{m=1}^{M}$ uniformly at random from subsets of $\{1, \ldots, N\} \setminus \{i\}$. Let $\pi \sim \Pi_M^{N,i}$ denote this sampling procedure with ($N-1$ choose $M$) possible subsets. This results in the objective function

$$O_2(\phi, \theta; X) \ = \ \sum_{i=1}^{N} \mathop{\mathbb{E}}_{\pi \sim \, \Pi_M^{N,i}} \log \sum_{m=1}^{M} \frac{1}{M} T_{\phi, \theta}(\mathbf{x}_i \mid \mathbf{x}_{\pi_m}) \,. \tag{11}$$

By moving $\mathbb{E}_\pi$ inside the log in (11) we recover $O_1$; *i.e.*, $O_2$ is a lower bound on $O_1$, via Jensen's inequality. Interestingly, we find $O_2$ often yields better generalization than $O_1$.

Once training is finished, all $N$ training exemplars are used to explain the generation of the validation or test sets using (1), for which the two regularizers discussed above are not used. Even though cross validation is commonly used for parameter tuning and model selection, in (11) cross validation is used as a training objective directly, suggestive of a *meta-learning* perspective. The non-parameteric nature of the exemplar based prior enables the use of the regularization techniques above, but this would not be straightforward for training parametric generative models.

**Learning objective.** To complete the definition of the learning objective for an Exemplar VAE, we combine RAT and exemplar sub-sampling to obtain the final Exemplar VAE objective:

$$O_3(\theta, \phi; X) = \sum_{i=1}^{N} \mathop{\mathbb{E}}_{q_\phi(\mathbf{z}|\mathbf{x}_i)} \left[ \log \frac{p_\theta(\mathbf{x}_i \mid \mathbf{z})}{q_\phi(\mathbf{z} \mid \mathbf{x}_i)} + \mathop{\mathbb{E}}_{\Pi_M^{N,i}(\pi)} \log \sum_{m=1}^{M} \frac{\mathbb{1}_{[\pi_m \in \mathrm{kNN}(\mathbf{z})]}}{(\sqrt{2\pi}\sigma)^{d_z}} \exp \frac{-\|\mathbf{z} - \boldsymbol{\mu}_\phi(\mathbf{x}_{\pi_m})\|^2}{2\sigma^2} \right],$$
$$\tag{12}$$

where, for brevity, the additive constant $-\log M$ has been dropped. We use the reparametrization trick to back propagate through $\mathbb{E}\, q_\phi(\mathbf{z} \mid \mathbf{x}_i)$. For small datasets and fully connected architectures we do not use RAT, but for convolutional models and large datasets the use of RAT is essential.

## 4   Related Work

Variational Autoencoders (VAEs) [32, 49] are versatile, latent variable generative models, used for non-linear dimensionality reduction [21], generating discrete data [5], and learning disentangled representations [26, 7], while providing a tractable lower bound on log marginal likelihood. Improved variants of the VAE are based on modifications to the VAE objective [6], more flexible variational familieis [33, 48], and more powerful decoders [8, 22]. More powerful latent priors [55, 2, 12, 35] can improve the effectiveness of VAEs for density estimation, as suggested by [27], and motivated by the observed gap between the prior and aggregated posterior (*e.g.,* [40]). More powerful priors may help avoid posterior collapse in VAEs with autoregressive decoders [5]. Unlike most existing work, Exemplar VAE assumes little about the structure of the latent space, using a non-parameteric prior.

VAEs with a VampPrior [55] optimize a set of pseudo-inputs together with the encoder network to obtain a Gaussian mixture approximation to the aggregate posterior. They argue that computing the exact aggregated posterior, while desirable, is expensive and suffers from overfitting, hence they restrict the number of pseudo-inputs to be much smaller than the training set. Exemplar VAE enjoys the use of all training points, but without a large increase in the the number of model parameters, while avoiding overfitting through simple regularization techniques. Training cost is reduced through RAT using approximate kNN search during training.

Exemplar VAE also extends naturally to large high dimensional datasets, and to discrete data, without requiring additional psedo-input parameters. VampPrior and Exemplar VAE are similar in their reuse of the encoder network and a mixture prior over the latent space. However, the encoder for the Exemplar VAE prior has a simplified covariance, which is useful for efficient learning. Importantly, we show that Exemplar VAEs can learn better unsupervised representations of images and perform generative data augmentation to improve supervised learning.

Memory augmented networks with attention can enhance generative models [36]. Hard attention has been used in VAEs [4] to generate images conditioned on memory items, with learnable and fixed memories. One can view Exemplar VAE as a VAE with external memory. One crucial difference between Exemplar VAE and [4] is in the conditional dependencies assumed in the Exemplar VAE, which disentangles the prior and reconstruction terms, and enables amortized computation per minibatch. In [4] discrete indices are optimized which creates challenges for gradient estimation, and they need to maintain a normalized categorical distribution over a potentially massive set of indices. By contrast, we use approximate kNN search in latent space to model hard attention, without requiring a normalized categorical distribution or high variance gradient estimates, and we mitigate overfitting using regularization.

Associative Compression Networsk [20] learn an ordering over a dataset to obtain better compression rates through VAEs. That work is similar to ours in defining the prior based on training data samples and the use of kNN in the latent space during training. However, their model with a conditional prior is not comparable with order agnostic VAEs. On the other hand, Exemplar VAE has an unconditional prior where, after training, defining an ordering is feasible and achieves the same goal

## 5  Experiments

**Experimental setup.**  We evaluate Exemplar VAE on density estimation, representation learning, and data augmentation. We use four datasets, namely, MNIST, Fashion-MNIST, Omniglot, and CelebA, and we consider four different architectures for gray-scale image data, namely, a VAE with MLP for encoder and decoder with two hidden layers (300 units each), a HVAE with similar architecture but two stochastic layers, ConvHVAE with two stochastic layers and convolutional encoder and decoder, and PixelSNAIL [9] with two stochastic layers and an auto-regressive PixelSNAIL shared between encoder and decoder. For CelebA we used a convolutional architecture based on [17]. We use gradient normalized Adam [31, 58] with learning rate 5e-4 and linear KL annealing for 100 epochs. See the supplementary material for details.

**Evaluation.**  For density estimation we use Importance Weighted Autoencoders (IWAE) [6] with 5000 samples, using the entire training set as exemplars, without regularization or kNN acceleration. This makes the evaluation time consuming, but generating an unbiased sample from the Exemplar VAE is efficient. Our preliminary experiments suggest that using kNN for evaluation is feasible.

### 5.1  Ablation Study

First, we evaluate the effectiveness of the regularization techniques proposed (Figure 2), *i.e.,* leave-one-out and exemplar subsampling, for enhancing generalization.

**Leave-one-out (LOO).**  We train an Exemplar VAE with a full aggregated exemplar based prior without RAT with and without LOO. Figure 2 plots the ELBO computed on training and validation sets, demonstrating the surprising effectiveness of LOO in regularization. Table 1 gives test log-likelihood IWAE bounds for Exemplar VAE on MNIST and Omniglot with and without LOO.

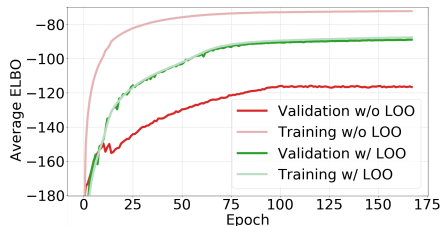

|          | Exemplar VAE | |
| Dataset  | w/ LOO   | w/o LOO   |
|----------|----------|-----------|
| MNIST    | $-82.35$ | $-101.33$ |
| Omniglot | $-105.80$ | $-139.12$ |

Figure 2: Training and validation ELBO on Dynamic MNIST for Exemplar VAE with and without LOO.

Table 1: Log likelihood lower bounds on the test set (nats) for Exemplar VAE with and without leave-one-out (LOO).

**Exemplar subsampling.**  As explained in Sec. 3.2, the Exemplar VAE uses a hyper-parameter $M$ to define the number of exemplars used for estimating the prior. Here, we report the Exemplar VAE's density estimates as a function of $M$ divided by the number of training data points $N$. We consider $M/N \in \{1.0, 0.5, 0.2, 0.1\}$. All models use LOO, and $M/N = 1$ reflects $M = N - 1$. Table 2 presents results for MNIST and Omniglot. In all of the following experiments we adopt $M/N = 0.5$.

| Dataset \ $M/N$ | 1 | 0.5 | 0.2 | 0.1 |
|----------|----------|----------|----------|----------|
| MNIST    | $-82.35$ | $\mathbf{-82.09}$ | $-82.12$ | $-82.20$ |
| Omniglot | $-105.80$ | $-105.22$ | $\mathbf{-104.95}$ | $-105.42$ |

Table 2: Test log likelihood lower bounds (nats) for Exemplar VAE versus fraction of exemplar subsampling.

### 5.2  Density Estimation

For each architecture, we compare to a Gaussian prior and a VampPrior, which represent the state-of-the-art among VAEs with a factored variational posterior. For training VAE and HVAE we did not

| Method | Dynamic MNIST | Fashion MNIST | Omniglot |
|---|---|---|---|
| VAE w/ Gaussian prior | $-84.45$ ±0.12 | $-228.70$ ±0.15 | $-108.34$ ±0.06 |
| VAE w/ VampPrior | $-82.43$ ±0.06 | $-227.35$ ±0.05 | $-106.78$ ±0.21 |
| Exemplar VAE | $\mathbf{-82.09}$ ±0.18 | $\mathbf{-226.75}$ ±0.07 | $\mathbf{-105.22}$ ±0.18 |
| HVAE w/ Gaussian prior | $-82.39$ ±0.11 | $227.37$ ±0.1 | $-104.92$ ±0.08 |
| HVAE w/ VampPrior | $-81.56$ ±0.09 | $-226.72$ ±0.08 | $-103.30$ ±0.43 |
| Exemplar HVAE | $\mathbf{-81.22}$ ±0.05 | $\mathbf{-226.53}$ ±0.09 | $\mathbf{-102.25}$ ±0.43 |
| ConvHVAE w/ Gaussian prior | $-80.52$ ±0.28 | $-225.38$ ±0.08 | $-98.12$ ±0.17 |
| ConvHVAE w/ Lars | $-80.30$ | $-225.92$ | $-97.08$ |
| ConvHVAE w/ SNIS | $-79.91$ ±0.05 | $-225.35$ ±0.07 | N/A |
| ConvHVAE w/ VampPrior | $-79.67$ ±0.09 | $-224.67$ ±0.03 | $-97.30$ ±0.07 |
| Exemplar ConvHVAE | $\mathbf{-79.58}$ ±0.07 | $\mathbf{-224.63}$ ±0.06 | $\mathbf{-96.38}$ ±0.24 |
| PixelSNAIL w/ Gaussian Prior | $-78.20$ ±0.01 | $-223.68$ ±0.01 | $-89.59$ ±0.04 |
| PixelSNAIL w/ VampPrior | $\mathbf{-77.90}$ ±0.01 | $-223.45$ ±0.01 | $-89.50$ ±0.05 |
| Exemplar PixelSNAIL | $-77.95$ ±0.01 | $\mathbf{-223.26}$ ±0.01 | $\mathbf{-89.28}$ ±0.06 |

Table 3: Density estimation on dynamic MNIST, Fashion MNIST, and Omniglot for different methods and architectures, all with 40-D latent spaces. Log likelihood lower bounds (nats), estimated with IWAE with 5000 samples, are averaged over 5 training runs. For LARS [2] and SNIS [35], the IWAE used 1000 samples; their architectures and training procedures are also somewhat different.

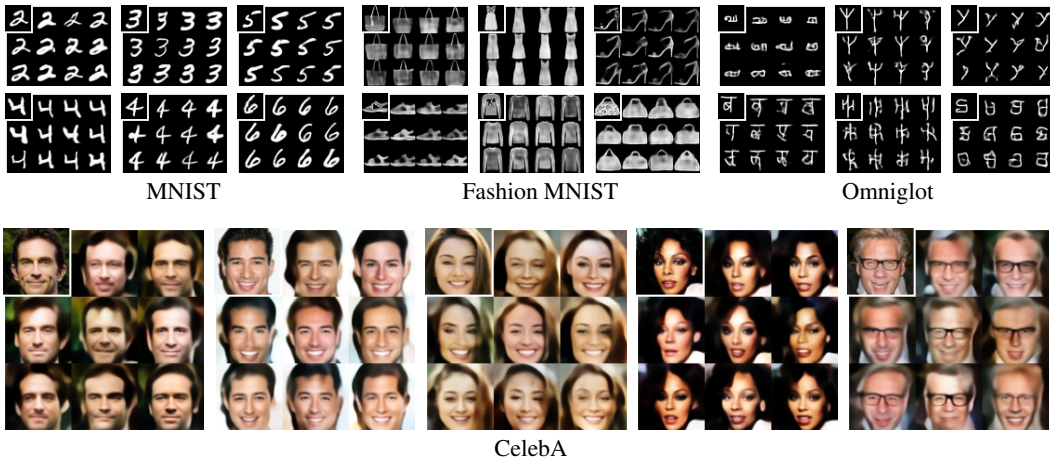

MNIST         Fashion MNIST         Omniglot

CelebA

Figure 3: Given a source exemplar on the top left of each plate, Exemplar VAE samples are generated, showing a significant diversity while preserving properties of the source exemplar.

utilize RAT, but for convolutional architectures we used RAT with 10NN search (see Sec. 3.1). Note that the number of nearest neighbors are selected based on computational budget; we believe larger values work better. Table 3 shows that Exemplar VAEs outperform other models in all cases except one. Improvement on Omniglot is greater than on other datasets, which may be due to its significant diversity. One can attempt to increase the number of pseudo-inputs in VampPrior, but this leads to overfitting. As such, we posit that Exemplar VAEs have the potential to more easily scale to large, diverse datasets. Note that training an Exemplar ConHVAE with approximate 10NN search is as efficient as training a ConHVAE with a VampPrior. Also, note that VampPrior [55] showed that a mixture of variational posteriors outperforms a Gaussian mixture prior, and hence we do not directly compare to that baseline.

Fig. 3 shows samples generated from an Exemplar ConvVAE, for which the corresponding exemplars are shown in the top left corner of each plate. These samples highlight the power of Exemplar VAE in maintaining the content of the source exemplar while adding diversity. For MNIST the changes are subtle, but for Fashion MNIST and Omniglot samples show more pronounced variation in style, possibly because those datasets are more diverse.

To assess the scalability of Exemplar VAEs to larger datasets, we train this model on $64 \times 64$ CelebA images [39]. Pixel values are modeled using a discretized logistic distribution [33, 51]. Exemplar

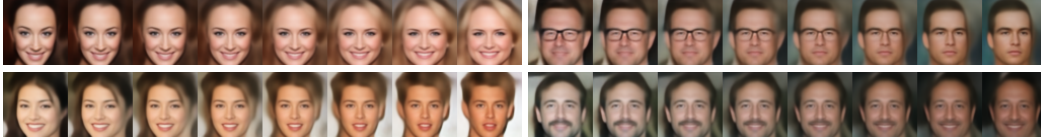

Figure 4: Interpolation between samples from the CelebA dataset.

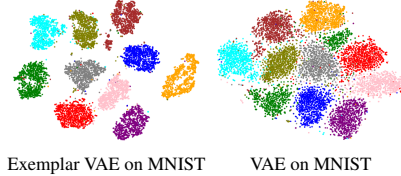

Exemplar VAE on MNIST    VAE on MNIST

Figure 5: t-SNE visualization of learned latent representations for test points, colored by labels.

| Method | MNIST | Fashion MNIST |
|---|---|---|
| VAE w/ Gaussian Prior | 2.41 ±0.27 | 15.90 ±0.34 |
| VAE w/ VampPrior | 1.42 ±0.02 | 12.74 ±0.18 |
| Exemplar VAE | **1.13** ±0.06 | **12.56** ±0.08 |

Table 4: kNN classification error (%) on 40-D unsupervised representations.

VAE samples (Figure 3) are high quality with good diversity. Interpolation in the latent space is also effective (Figure 4). More details and quantitative evaluations are provided in the supplementary materials.

## 5.3 Representation Learning

We next explore the structure of the latent representation for Exemplar VAE. Fig. 5 shows a t-SNE visualization of the latent representations of MNIST test data for the Exemaplar VAE and for VAE with a Gaussian prior. Test points are colored by their digit label. No labels were used during training. The Exemplar VAE representation appears more meaningful, with tighter clusters than VAE. We also use k-nearest neighbor (kNN) classification performance as a proxy for the representation quality. As is clear from Table 4, Exemplar VAE consistently outperforms other approaches. Results on Omniglot are not reported since the low resolution variant of this dataset does not include class labels. We also counted the number of active dimension in the latent to measure posterior collapse. Section D of supplementary materials shows the superior behavior of Exemplar VAE.

## 5.4 Generative Data Augmentation

Finally, we ask whether Exemplar VAE is effective in generating augmented data to improve supervised learning. Recent generative models have achieved impressive sample quality and diversity, but limited success in improving discriminative models. Class-conditional models were used to generate training data, but with marginal gains [46]. Techniques for optimizing geometric augmentation policies [11, 38, 24] and adversarial perturbations [19, 41] were more successful for classification.

Here we use the original training data as exemplars, generating extra samples from Exemplar VAE. Class labels of source exemplars are transferred to corresponding generated images, and a combination of real and generated data is used for supervised learning. Each training iteration involves 3 steps:

1. Draw a minibatch $X = \{(\mathbf{x}_i,\ y_i)\}_{i=1}^{B}$ from training data.
2. For each $\mathbf{x}_i \in X$, draw $\mathbf{z}_i \sim r_\phi(\mathbf{z} \mid \mathbf{x}_i)$, and then set $\tilde{\mathbf{x}}_i = \boldsymbol{\mu}_\phi(\mathbf{x} \mid \mathbf{z}_i)$, which inherits the class label $y_i$. This yields a synthetic minibatch $\tilde{X} = \{(\tilde{\mathbf{x}}_i,\ y_i)\}_{i=1}^{B}$.
3. Optimize the weighted cross entropy: $\ell = -\sum_{i=1}^{B}\left[\lambda \log p_\theta(y_i | \mathbf{x}_i) + (1-\lambda)\log p_\theta(y_i | \tilde{\mathbf{x}}_i)\right]$

For VAE with Gaussian prior and VampPrior we sampled from variational posterior instead of $r_\phi$. We train MLPs with ReLU activations and two hidden layers of 1024 or 8192 units on MNIST and Fashion MNIST. We leverage label smoothing [54] with a parameter of $0.1$. The Exemplar VAEs used for data augmentation have fully connected layers and are not trained with class labels.

Fig. 6 shows Exemplar VAE is more effective than other VAEs for data augmentation. Even small amounts of generative data augmentation improves classifier accuracy. A classifier trained solely on synthetic data achieves better error rates than one trained on the original data. Given $\lambda = 0.4$ on MNIST and $\lambda = 0.8$ on Fashion MNIST, we train 10 networks on the union of training and validation sets and report average test errors. On permutation invariant MNIST, Exemplar VAE augmentations achieve an average error rate of $0.69\%$. Tables 5 and 6 summarize the results in comparison with

previous work. Ladder Networks [52] and Virtual Adversarial Training [41] report error rates of $0.57\%$ and $0.64\%$ on MNIST, using deeper architectures and more complex training procedures.

| Method | Hidden layers | Test error |
|---|---|---|
| Dropout [53] | $3 \times 1024$ | 1.25 |
| Label smoothing [45] | $2 \times 1024$ | $1.23 \pm 0.06$ |
| Dropconnect [56] | $2 \times 800$ | 1.20 |
| VIB [1] | $2 \times 1024$ | 1.13 |
| Dropout + MaxNorm [53] | $2 \times 8192$ | 0.95 |
| MTC [50] | $2 \times 2000$ | 0.81 |
| DBM + DO fine. [53] | $500\text{-}500\text{-}2K$ | 0.79 |
| Label Smoothing (LS) | $2 \times 1024$ | $1.23 \pm 0.01$ |
| LS+Exemplar VAE Aug. | $2 \times 1024$ | $0.77 \pm 0.01$ |
| Label Smoothing | $2 \times 8196$ | $1.17 \pm 0.01$ |
| LS+Exemplar VAE Aug. | $2 \times 8192$ | $\mathbf{0.69} \pm 0.01$ |

Table 5: Test error (%) on permutation invariant MNIST from [53, 45, 56, 1, 50], and our results with and without generative data augmentation.

| Method | Hidden layers | Test error |
|---|---|---|
| Label Smoothing | $2 \times 1024$ | $8.96 \pm 0.04$ |
| LS+Exemplar VAE Aug. | $2 \times 1024$ | $8.46 \pm 0.04$ |
| Label Smoothing | $2 \times 8196$ | $8.56 \pm 0.03$ |
| LS+Exemplar VAE Aug. | $2 \times 8192$ | $\mathbf{8.16} \pm 0.03$ |

Table 6: Test error (%) on permutaion invariant Fashion MNIST.

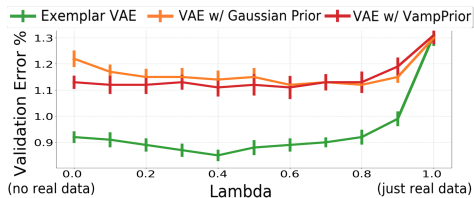

Figure 6: MNIST validation error versus $\lambda$, which controls the relative balance of real and augmented data, for different generative models.

# 6 Conclusion

We develop a framework for exemplar based generative modeling called the Exemplar VAE. We present two effective regularization techniques for Exemplar VAEs, and an efficient learning algorithm based on approximate nearest neighbor search. The effectiveness of the Exemplar VAE on density estimation, representation learning, and data augmentation for supervised learning is demonstrated. The development of Exemplar VAEs opens up interesting future research directions such as application to NLP (cf. [23]) and other discrete data, further exploration of unsupervised data augmentation, and extentions to other generative models such as Normalizing Flows and GANs.

## Broader Impact Statement

The ideas described in our paper concern the development of a new fundamental class of unsupervised learning algorithm, rather than an application per se. One important property of the method stems from it's non-parametric form, i.e., as an exemplar-based model. As such, rather than having the "model" represented solely in the weights of an amorphous non-linear neural network, in our case much of the model is expressed directly in terms of the dataset of exemplars. As such, the model is somewhat more interpretable and may facilitate the examination or discovery of bias, which has natural social and ethical implications. Beyond that, the primary social and ethical implications will derive from the way in which the algorithm is applied in different domains.

## Funding Disclosure

This research was supported in part by an NSERC Discovery Grant to DJF, and by Province of Ontario, the Government of Canada, through NSERC and CIFAR, and companies sponsoring the Vector Institute.

## Acknowledgement

We are extremely grateful to Micha Livne, Will Grathwohl, and Kevin Swersky for extensive discussions. We thank Alireza Makhzani, Kevin Murphy, Abhishek Gupta, and Alex Alemi for useful discussions and Diederik Kingma, Chen Li, Danijar Hafner, and David Duvenaud for their valuable feedback on an initial draft of this paper.

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
