[Supplementary Material]

# Supplementary Materials for Exemplar VAE: Linking Generative Models, Nearest Neighbor Retrieval, and Data Augmentation

## A  Exemplar VAE samples

MNIST

Fashion MNIST

Omniglot

CelebA

Figure 1: Random samples drawn from Exemplar VAEs trained on different datasets.

# B   Exemplar conditioned samples

Figure 2: Given the input exemplar on the top left of each plate, 11 exemplar conditioned samples using Exemplar VAE are generated and shown.

## C Retrieval Augmented Training

**Algorithm 1**

---

**Input:** Training dataset $\mathcal{X} = \{\mathbf{x}_n\}_{n=1}^N$
**Define** Cache:
    initialize cache = []
    insert($i, \boldsymbol{c}$): insert value $\boldsymbol{c}$ with index $i$ into cache
    update($i, \boldsymbol{c}$): update the value of index $i$ to $\boldsymbol{c}$
    kNN($\boldsymbol{c}$): return indices of kNNs of $\boldsymbol{c}$ in cache
**for** $n$ **in** $\{1, \ldots, N\}$ **do** Cache.insert($n, \boldsymbol{\mu}_\phi(\mathbf{x}_n)$)
**for** epoch **in** $\{1, \ldots, L\}$ **do**
  **for** $i$ **in** $\{1, \ldots, N\}$ **do**
    $\pi \sim \Pi_M^{N,i}$ to obtain a set of $M$ exemplar indices
    $\boldsymbol{\mu}_i, \Lambda_i = \boldsymbol{\mu}_\phi(\mathbf{x}_i), \Lambda_\phi(\mathbf{x}_i)$
    $\boldsymbol{\epsilon} \sim \mathcal{N}(0, I_{d_z \times d_z})$
    $\mathbf{z} = \boldsymbol{\mu}_i + \Lambda_i^{1/2} \boldsymbol{\epsilon}$
    kNN = Cache.kNN($\boldsymbol{\mu}_i$) $\cap$ $\pi$
    **for** $j$ **in** kNN **do** $\boldsymbol{\mu}_j = \boldsymbol{\mu}_\phi(\mathbf{x}_j)$
    $m(\mathbf{z}) = \frac{1}{M} \sum_{j \in \text{kNN}} \mathcal{N}(\mathbf{z} \,|\, \boldsymbol{\mu}_j, \sigma^2)$
    $\text{ELBO} = \log p_\theta(\mathbf{x} \,|\, \mathbf{z}) - \log \mathcal{N}(\mathbf{z} \,|\, \boldsymbol{\mu}_i, \Lambda_i) + \log r(\mathbf{z})$
    Gradient ascend on ELBO to update $\phi, \theta$, and $\sigma^2$
    Cache.update($i, \boldsymbol{\mu}_i$)
    **for** $j$ **in** kNN **do** Cache.update($j, \boldsymbol{\mu}_j$)

---

## D Number of Active Dimensions in the Latent Space

The problem of posterior collapse [2, 5], resulting in a number of inactive dimensions in the latent space of a VAE. We investigate this phenomena by counting the number of active dimensions based on a metric proposed by Burda et. al [3]. This metric computes the variance of the mean of the latent encoding of the data points in each dimension of the latent space, $\text{Var}(\mu_\phi(\mathbf{x})_i)$, where $\mathbf{x}$ is sampled from the dataset. If the computed variance is above a certain threshold, then that dimension is considered active. The proposed threshold by [1] is 0.01 and we use the same value. We observe that the Exemplar VAE has the largest number of active dimensions in all cases except one. In the case of ConvHVAE on MNIST and Fashion MNIST, the gap between Exemplar VAE and other methods is more considerable.

| Model | Number of active dimensions out of 40 | | |
| --- | --- | --- | --- |
| | Dynamic MNIST | Fashion MNIST | Omniglot |
| VAE w/ Gaussian prior | 24.0±0.63 | 26.0±1.1 | 35.2±0.4 |
| VAE w/ Vampprior | 27.6±1.36 | 35.25±1.3 | **40.0**±0.0 |
| Exemplar VAE | **29.4**±0.49 | **36.0**±1.41 | **40.0**±0.0 |
| HVAE w/ Gaussian prior | 15.0±0.63 | 12.4±0.8 | 24.8±1.83 |
| HVAE w/ VampPrior | 20.4±0.49 | 23.2±1.47 | **39.0**±0.89 |
| Exemplar HVAE | **21.6**±0.49 | **28.6**±0.8 | 38.6±1.5 |
| ConvHVAE w/ Gaussian prior | 19.8±2.93 | 15.4±2.65 | 39.2±1.6 |
| ConvHVAE w/ VampPrior | 19.0±1.55 | 19.25±0.83 | 39.8±0.4 |
| Exemplar ConvHVAE | **25.8**±3.66 | **33.6**±7.86 | **40.0**±0.0 |

Table 1: The number of active dimensions computed based on a metric proposed by Burda et. al [3]. This metric considers a latent dimension active if the variance of its mean over the dataset is higher than 0.01. For hierarchical architectures the reported number is for the $\mathbf{z}_2$ which is the highest stochastic layer.

# E  CelebA Quantitative Results

| Model | bits per dim |
|---|---|
| VAE w/ Gaussian Prior | 5.825 |
| Exemplar VAE | **5.780** |

Table 2: Numerical Evaluations for CelebA

# F  Derivation of Eqn. (5)

$$\log p(\mathbf{x}; X, \theta, \phi) = \log \sum_{n=1}^{N} \frac{1}{N} \int_z r_\phi(\mathbf{z} \mid \mathbf{x}_n) \, p_\theta(\mathbf{x} \mid \mathbf{z}) \, d\mathbf{z} \tag{1}$$

$$= \log \int_z p_\theta(\mathbf{x} \mid \mathbf{z}) \sum_{n=1}^{N} \frac{1}{N} r_\phi(\mathbf{z} \mid \mathbf{x}_n) \, d\mathbf{z} \tag{2}$$

$$= \log \int_z \frac{q_\phi(\mathbf{z} \mid \mathbf{x}) p_\theta(\mathbf{x} \mid \mathbf{z}) \sum_{n=1}^{N} \frac{1}{N} r_\phi(\mathbf{z} \mid \mathbf{x}_n)}{q_\phi(\mathbf{z}|\mathbf{x})} \, d\mathbf{z} \tag{3}$$

$$\geq \underbrace{\mathbb{E}_{q_\phi(\mathbf{z}|\mathbf{x})} \log p_\theta(\mathbf{x}|\mathbf{z})}_{\text{reconstruction}} - \underbrace{\mathbb{E}_{q_\phi(\mathbf{z}|\mathbf{x})} \log \frac{q_\phi(\mathbf{z} \mid \mathbf{x})}{\sum_{n=1}^{N} r_\phi(\mathbf{z} \mid \mathbf{x}_n)/N}}_{\text{KL term}} \tag{4}$$

$$= O(\theta, \phi; \mathbf{x}, X). \tag{5}$$

# G  Iterative generation

The exemplar VAE generates a new sample by stochastically transforming an exemplar. The newly generated data point can also be used as an exemplar, and we can repeat this procedure again and again. This kind of generation bears some similarity to MCMC for sampling from energy-based models. Figure 3 shows how samples evolve and consistently stay near the manifold of MNIST digits. We can apply the same procedure starting from a noisy input image as an exemplar. Figure 4 shows that the model is able to quickly transform the noisy images into samples that resemble real MNIST images.

Figure 3: Iterative generation starting from a training data point. Samples generated from an Exemplar VAE starting from a training data point, and then reusing the generated data as exemplars for the next round of generation (left to right).

Figure 4: Iterative generation starting from a noise input (left to right).

## H Computation and Memory Complexity

The cost of training Exemplar VAE is similar to that of VampPrior, which uses mixture of variational posteriors. When the number of exemplars per minibatch is equal to the number of pseudo-inputs in VampPrior the computational complexity is very similar. For example, for ConvHVAE on Omniglot, VampPrior with 1000 pseudo-inputs takes 58s/epoch and Exemplar VAE with a minibatch of 100 and 10 NNs takes 51s/epoch on a single Nvidia T4 GPU (it runs faster because we use an isotropic gaussians in our prior). In case of ConvHVAE on MNIST and FashionMNIST VampPrior with 500 pseudo inputs takes 82s/epoch vs 107s/epoch for Exemplar VAE with batch size of 100 and 10 NNs per data point. Regarding memory complexity, Exemplar VAE stores low-dimensional latent embeddings. By comparison, VampPrior stores pseudo inputs with the same dimentionality as the input data, which can be problematic in case of high dimensional data.

## I Reconstruction vs. KL

Table 3 shows the value of KL and the reconstruction terms of ELBO, computed based on a single sample from the variational posterior, averaged across test set. These numbers show that not only the exemplar VAE improves the KL term, but also the reconstruction terms are comparable with the VampPrior.

| | Dynamic MNIST | | Fashion MNIST | | Omniglot | |
|---|---|---|---|---|---|---|
| Model | KL | Neg.Reconst. | KL | Neg. Reconst. | KL | Neg. Reconst. |
| VAE w/ Gaussian prior | $25.54\pm0.12$ | $63.06\pm0.11$ | $18.38\pm0.11$ | $213.21\pm0.18$ | $32.97\pm0.2$ | $82.3\pm0.21$ |
| VAE w/ VampPrior | $25.14\pm0.16$ | $\mathbf{60.79}\pm0.13$ | $18.44\pm0.06$ | $211.37\pm0.04$ | $34.17\pm0.22$ | $\mathbf{79.49}\pm0.18$ |
| Exemplar VAE | $\mathbf{24.82}\pm0.22$ | $61.00\pm0.13$ | $\mathbf{18.32}\pm0.08$ | $\mathbf{211.10}\pm0.1$ | $\mathbf{32.66}\pm0.27$ | $80.25\pm0.62$ |
| HVAE w/ Gaussian prior | $26.80\pm0.13$ | $59.80\pm0.11$ | $19.08\pm0.05$ | $211.18\pm0.14$ | $\mathbf{36.07}\pm0.12$ | $75.96\pm0.12$ |
| HVAE w/ VampPrior | $26.69\pm0.1$ | $\mathbf{58.46}\pm0.06$ | $19.27\pm0.15$ | $\mathbf{210.04}\pm0.2$ | $38.39\pm0.16$ | $\mathbf{72.42}\pm0.34$ |
| Exemplar HVAE | $\mathbf{26.41}\pm0.17$ | $58.48\pm0.16$ | $\mathbf{18.96}\pm0.15$ | $210.40\pm0.16$ | $36.76\pm0.25$ | $73.35\pm0.63$ |
| ConvHVAE w/ Gaussian prior | $26.58\pm0.27$ | $57.64\pm0.57$ | $20.34\pm0.04$ | $208.11\pm0.06$ | $38.90\pm0.22$ | $67.22\pm0.1$ |
| ConvHVAE w/ VampPrior | $26.57\pm0.17$ | $56.18\pm0.03$ | $20.65\pm0.19$ | $\mathbf{206.64}\pm0.15$ | $38.95\pm0.17$ | $\mathbf{66.38}\pm0.3$ |
| Exemplar ConvHVAE | $\mathbf{26.41}\pm0.25$ | $\mathbf{56.14}\pm0.27$ | $\mathbf{20.46}\pm0.23$ | $207.18\pm0.38$ | $\mathbf{37.48}\pm0.37$ | $66.62\pm0.32$ |

Table 3: KL and reconstruction part of ELBO averaged over test set by a single sample from posterior.

## J t-SNE visualization of Fashion MNIST latent space

We showed t-SNE visualization of MNIST latent space in the figure 5. Here we show the same plot for fashion-mnist. Interestingly, some classes are very close to each other (Pullover-shirt-dress) and transition between them happens very smoothly while some other classes are more separated.

Exemplar VAE on Fashion MNIST          VAE on Fashion MNIST

Table 4: t-SNE visualization of learned latent representations for Fashion-MNIST test points, colored by labels.

# K   Sub-sampling for VampPrior

To regularize the Exemplar VAE, we used leave-one-out and exemplar sub-sampling. The use of leave-one-out is enabled by the non-parametric nature of the prior. It is not clear how to apply the same regularization to VampPrior, but it is possible to apply mixture component sub-sampling for VampPrior as well. VampPrior showed that it outperforms a VAE with a mixture of Gaussians prior. That is why did not compare directly against a mixture model prior in the primary experimental section. Here we explore applying subsampling to VampPrior and confirm that mixture of Gaussian prior exhibit inferior performance when compared to both VampPrior and the exemplar based Prior.

| Model | Log-Likelihood |
|---|---|
| VAE w/ Gaussian Prior | $-108.34$ |
| VAE w/ Mixture of Gaussians Prior | $-107.49$ |
| VAE w/ VampPrior w/o sub-sampling, components=1000 | $-106.78$ |
| VAE w/ VampPrior w/ sub-sampling, components=1000 | $-106.24$ |
| VAE w/ VampPrior w/ sub-sampling, components=2000 | $-106.16$ |
| VAE w/ VampPrior w/ sub-sampling, components=5000 | $-106.37$ |
| VAE w/ Exemplar Prior w/ sub-sampling, components=11500 | $-105.22$ |

Table 5: Test set log-likelihood numbers on Omniglot for different setups with or without sub-sampling

While the subsampling of the mixture components helps the performance of vampprior, but still outperformed by exemplar prior.

# L   Experimental Details

## L.1   Architectures

All of the neural network architectures are based on the VampPrior of Tomczak & Welling [6][1] except PixelSNAIL. We leave tuning the architecture of Exemplar VAEs to future work. To describe the network architectures, we follow the notation of LARS [1]. Neural network layers used are either convolutional (denoted CNN) or fully-connected (denoted MLP), and the number of units are written inside a bracket separated by a dash (e.g., MLP[300-784] means a fully-connected layer with 300 input units and 784 output units). We use curly bracket to show concatenation.

Three different architectures are used in the experiments, described below. $d_z$ refers to the dimensionality of the latent space.

a) **VAE**:

$$
\begin{aligned}
q_\phi(\mathbf{z} \mid \mathbf{x}) &= \mathcal{N}(\mathbf{z};\ \mu_{\mathbf{z}}(\mathbf{x}), \Lambda_{\mathbf{z}}(\mathbf{x})) \\
p_\phi(\mathbf{x} \mid \mathbf{z}) &= \mathrm{Bernoulli}(x, \mu_{\mathbf{x}}(\mathbf{z}))
\end{aligned}
$$

$$
\begin{aligned}
\mathrm{Encoder}_{\mathbf{z}}(\mathbf{x}) &= \mathrm{MLP}\,[784 - 300 - 300] \\
\log \Lambda_{\mathbf{z}}^2(\mathbf{x}) &= \mathrm{MLP}[\mathrm{Encoder}_{\mathbf{z}}(x) - d_{\mathbf{z}}] \\
\mu_{\mathbf{z}}(\mathbf{x}) &= \mathrm{MLP}[\mathrm{Encoder}_{\mathbf{z}}(x) - d_{\mathbf{z}}] \\
\mu_{\mathbf{x}}(\mathbf{z}) &= \mathrm{MLP}[d_{\mathbf{z}} - 300 - 300 - 784]
\end{aligned}
$$

b) **HVAE**:

$$
\begin{aligned}
q_\phi(\mathbf{z}_2 \mid \mathbf{x}) &= \mathcal{N}(\mathbf{z}_2;\ \mu_{\mathbf{z}_2}(\mathbf{x}), \Lambda_{\mathbf{z}_2}(\mathbf{x})) \\
q_\phi(\mathbf{z}_1 \mid \mathbf{x},\ \mathbf{z}_2) &= \mathcal{N}(\mathbf{z}_1;\ \mu_{\mathbf{z}_1}(\mathbf{x},\ \mathbf{z}_2), \Lambda_{\mathbf{z}_1}(\mathbf{x},\ \mathbf{z}_2)) \\
p_\phi(\mathbf{z}_1 \mid \mathbf{z}_2) &= \mathcal{N}(\mathbf{z}_1;\ \hat{\mu}_{\mathbf{z}_1}(\mathbf{z}_2), \hat{\Lambda}_{\mathbf{z}_1}(\mathbf{z}_2)) \\
p_\phi(\mathbf{x} \mid \mathbf{z}_1,\ \mathbf{z}_2) &= \mathrm{Bernoulli}(\mathbf{x}, \mu_{\mathbf{x}}(\mathbf{z}_1,\ \mathbf{z}_2))
\end{aligned}
$$

$$
\begin{aligned}
\mathrm{Encoder}_{\mathbf{z}_2}(\mathbf{x}) &= \mathrm{MLP}[784 - 300 - 300] \\
\log \Lambda_{\mathbf{z}_2}^2(\mathbf{x}) &= \mathrm{MLP}[\mathrm{Encoder}_{\mathbf{z}_2}(\mathbf{x}) - d_{\mathbf{z}_2}] \\
\mu_{\mathbf{z}_2}(\mathbf{x}) &= \mathrm{MLP}[\mathrm{Encoder}_{\mathbf{z}_2}(\mathbf{x}) - d_{\mathbf{z}_2}] \\
\mathrm{Encoder}_{\mathbf{z}_1}(\mathbf{x}, \mathbf{z}_2) &= \mathrm{MLP}[\{\mathrm{MLP}[d_{\mathbf{z}_2} - 300], \mathrm{MLP}[784 - 300]\} - 300] \\
\log \Lambda_{\mathbf{z}_1}^2(\mathbf{x}, \mathbf{z}_2) &= \mathrm{MLP}[\mathrm{Encoder}_{\mathbf{z}_1}(\mathbf{x}, \mathbf{z}_2) - d_{\mathbf{z}_1}] \\
\mu_{\mathbf{z}_1}(\mathbf{x}, \mathbf{z}_2) &= \mathrm{MLP}[\mathrm{Encoder}_{\mathbf{z}_1}(\mathbf{x}, \mathbf{z}_2) - d_{\mathbf{z}_1}] \\
\mathrm{Decoder}_{\mathbf{z}_1}(\mathbf{z}_2) &= \mathrm{MLP}[d_{\mathbf{z}_2} - 300 - 300] \\
\log \hat{\Lambda}_{\mathbf{z}_1}^2(\mathbf{z}_2) &= \mathrm{MLP}[\mathrm{Decoder}_{\mathbf{z}_1}(\mathbf{z}_2) - d_{\mathbf{z}_1}] \\
\hat{\mu}_{\mathbf{z}_1}(\mathbf{z}_2) &= \mathrm{MLP}[\mathrm{Decoder}_{\mathbf{z}_1}(\mathbf{z}_2) - d_{\mathbf{z}_1}] \\
\mu_{\mathbf{x}}(\mathbf{z}_1, \mathbf{z}_2) &= MLP[\{\mathrm{MLP}[d_{\mathbf{z}_1} - 300], \mathrm{MLP}[d_{\mathbf{z}_2} - 300]\} - 300 - 784]
\end{aligned}
$$

c) **ConvHVAE**: The generative and variational posterior distributions are identical to HVAE.

$$
\begin{aligned}
\mathrm{Encoder}_{\mathbf{z}_2}(\mathbf{x}) &= \mathrm{CNN}[28 \times 28 \times 1 - 32 \times 32 \times 32 - 12 \times 12 \times 32 - 12 \times 12 \times 64 - 7 \times 7 \times 64 \\
&\quad -7 \times 7 \times 6] \\
\log \Lambda_{\mathbf{z}_2}^2(\mathbf{x}) &= \mathrm{MLP}[\mathrm{Encoder}_{\mathbf{z}_2}(\mathbf{x}) - d_{\mathbf{z}_2}] \\
\mu_{\mathbf{z}_2}(\mathbf{x}) &= \mathrm{MLP}[\mathrm{Encoder}_{\mathbf{z}_2}(\mathbf{x}) - d_{\mathbf{z}_2}] \\
\mathrm{ConvEncoder}_{\mathbf{z}_1}(\mathbf{x}) &= \mathrm{CNN}[28 \times 28 \times 1 - 32 \times 32 \times 32 - 12 \times 12 \times 32 - 12 \times 12 \times 64 - 7 \times 7 \times 64 - 7 \times 7 \times 6] \\
\mathrm{Encoder}_{\mathbf{z}_1}(\mathbf{x}, \mathbf{z}_2) &= \mathrm{MLP}[\{\mathrm{MLP}[d_{\mathbf{z}_2} \\
&\quad -7 \times 7 \times 6], \mathrm{ConvEncoder}_{\mathbf{z}_1}(\mathbf{x})\} - 300] \\
\log \Lambda_{\mathbf{z}_1}^2(\mathbf{x}, \mathbf{z}_2) &= \mathrm{MLP}[\mathrm{Encoder}_{\mathbf{z}_1}(\mathbf{x}, \mathbf{z}_2) - d_{\mathbf{z}_1}] \\
\mu_{\mathbf{z}_1}(\mathbf{x}, \mathbf{z}_2) &= \mathrm{MLP}[\mathrm{Encoder}_{\mathbf{z}_1}(\mathbf{x}, \mathbf{z}_2) - d_{\mathbf{z}_1}] \\
\mathrm{Decoder}_{\mathbf{z}_1}(\mathbf{z}_2) &= \mathrm{MLP}[d_{\mathbf{z}_2} - 300 - 300] \\
\log \hat{\Lambda}_{\mathbf{z}_1}^2(\mathbf{z}_2) &= \mathrm{MLP}[\mathrm{Decoder}_{\mathbf{z}_1}(\mathbf{z}_2) - d_{\mathbf{z}_1}] \\
\hat{\mu}_{\mathbf{z}_1}(\mathbf{z}_2) &= \mathrm{MLP}[\mathrm{Decoder}_{\mathbf{z}_1}(\mathbf{z}_2) - d_{\mathbf{z}_1}] \\
\mathrm{MLPDecoder}_{\mathbf{x}}(\mathbf{z}_1, \mathbf{z}_2) &= \mathrm{MLP}[\{\mathrm{MLP}[d_{\mathbf{z}_1} - 300], \mathrm{MLP}[d_{\mathbf{z}_2} - 300]\} - 784] \\
\mathrm{ConvDecoder}_{\mathbf{x}} &= \mathrm{CNN}[28 \times 28 \times 64 - 28 \times 28 \times 64 - 28 \times 28 \times 64 - 28 \times 28 \times 64 - 28 \times 28 \times 1] \\
\mu_{\mathbf{x}}(\mathbf{z}_1, \mathbf{z}_2) &= [\mathrm{MLPDecoder}_{\mathbf{x}}(\mathbf{z}_1, \mathbf{z}_2) - \mathrm{ConvDecoder}_{\mathbf{x}}]
\end{aligned}
$$

d) **PixelSNAIL HVAE**: The generative and variational posterior distributions are identical to HVAE.

$$
\begin{aligned}
\text{Encoder}_{\mathbf{z}_2}(\mathbf{x}) &= \text{CNN}[28 \times 28 \times 1 - 32 \times 32 \times 32 - 12 \times 12 \times 32 - 12 \times 12 \times 64 \\
&\quad -7 \times 7 \times 64 - 7 \times 7 \times 6] \\
\log \Lambda_{\mathbf{z}_2}^2(\mathbf{x}) &= \text{MLP}[\text{Encoder}_{\mathbf{z}_2}(\mathbf{x}) - d_{\mathbf{z}_2}] \\
\mu_{\mathbf{z}_2}(\mathbf{x}) &= \text{MLP}[\text{Encoder}_{\mathbf{z}_2}(\mathbf{x}) - d_{\mathbf{z}_2}] \\
\text{ConvEncoder}_{\mathbf{z}_1}(\mathbf{x}) &= \text{CNN}[28 \times 28 \times 1 - 32 \times 32 \times 32 - 12 \times 12 \times 32 - 12 \times 12 \times 64 \\
&\quad -7 \times 7 \times 64 - 7 \times 7 \times 6] \\
\text{Encoder}_{\mathbf{z}_1}(\mathbf{x}, \mathbf{z}_2) &= \text{MLP}[\{\text{MLP}[d_{\mathbf{z}_2} - 7 \times 7 \times 6], \text{ConvEncoder}_{\mathbf{z}_1}(\mathbf{x})\} - 300] \\
\log \Lambda_{\mathbf{z}_1}^2(\mathbf{x}, \mathbf{z}_2) &= \text{MLP}[\text{Encoder}_{\mathbf{z}_1}(\mathbf{x}, \mathbf{z}_2) - d_{\mathbf{z}_1}] \\
\mu_{\mathbf{z}_1}(\mathbf{x}, \mathbf{z}_2) &= \text{MLP}[\text{Encoder}_{\mathbf{z}_1}(\mathbf{x}, \mathbf{z}_2) - d_{\mathbf{z}_1}] \\
\text{Decoder}_{\mathbf{z}_1}(\mathbf{z}_2) &= \text{MLP}[d_{\mathbf{z}_2} - 300 - 300] \\
\log \hat{\Lambda}_{\mathbf{z}_1}^2(\mathbf{z}_2) &= \text{MLP}[\text{Decoder}_{\mathbf{z}_1}(\mathbf{z}_2) - d_{\mathbf{z}_1}] \\
\hat{\mu}_{\mathbf{z}_1}(\mathbf{z}_2) &= \text{MLP}[\text{Decoder}_{\mathbf{z}_1}(\mathbf{z}_2) - d_{\mathbf{z}_1}] \\
\text{MLPDecoder}_{\mathbf{x}}(\mathbf{z}_1, \mathbf{z}_2, \mathbf{x}) &= \{\text{MLP}[d_{\mathbf{z}_1} - 784], \text{MLP}[d_{\mathbf{z}_2} - 784], \mathbf{x}\} \\
\text{AutoRegressiveDecoder}_{\mathbf{x}} &= [\text{MaskedCNN}[28 \times 28 \times 64 - 28 \times 28 \times 64 - 28 \times 28 \times 64 - 28 \times 28 \times 64] \\
&\quad -\text{Self-Attention} - \text{MaskedCNN}[28 \times 28 \times 1]] \\
\mu_{\mathbf{x}}(\mathbf{z}_1, \mathbf{z}_2) &= [\text{MLPDecoder}_{\mathbf{x}}(\mathbf{z}_1, \mathbf{z}_2, \mathbf{x}) - \text{AutoRegressiveDecoder}_{\mathbf{x}}]
\end{aligned}
$$

e) **CelebA Architecture**:

$$
\begin{aligned}
q_\phi(\mathbf{z} \mid \mathbf{x}) &= \mathcal{N}(\mathbf{z};\ \mu_{\mathbf{z}}(\mathbf{x}), \Lambda_{\mathbf{z}}(\mathbf{x})) \\
p_\phi(\mathbf{x} \mid \mathbf{z}) &= \text{Discretized\_Logistic}(x, \mu_{\mathbf{x}}(\mathbf{z}), \sigma^2)
\end{aligned}
$$

$$
\begin{aligned}
\text{Encoder}_{\mathbf{z}}(\mathbf{x}) &= \text{CNN}\,[64 \times 64 \times 3 - 32 \times 32 \times 64 - 16 \times 16 \times 128 - 8 \times 8 \times 256 - 4 \times 4 \times 512] \\
\log \Lambda_{\mathbf{z}}^2(\mathbf{x}) &= \text{MLP}[\text{Encoder}_{\mathbf{z}}(x) - d_{\mathbf{z}}] \\
\mu_{\mathbf{z}}(\mathbf{x}) &= \text{MLP}[\text{Encoder}_{\mathbf{z}}(x) - d_{\mathbf{z}}] \\
\mu_{\mathbf{x}}(\mathbf{z}) &= \text{CNN}[8 \times 8 \times 512 - 16 \times 16 \times 256 - 32 \times 32 \times 128 - 64 \times 64 \times 64 - 64 \times 64 \times 3]
\end{aligned}
$$

As the activation function, the gating mechanism of [4] is used throughout. So for each layer we have two parallel branches where the sigmoid of one branch is multiplied by the output of the other branch. In ConvHVAE the kernel size of the first layer of $\text{Encoder}_{\mathbf{z}_2}(x)$ is 7 and the third layer used kernel size of 5. The last layer of $\text{ConvDecoder}_{\mathbf{x}}$ used kernel size of 1 and all the other layers used $3 \times 3$ kernels. For CelebA we used kernel size of 5 for each layer and combination of batch norm and ELU activation after each convolution layer.

## L.2 Hyper-parameters

We use Graident Normalized Adam [7] with Learning rate of $5e - 4$ and minibatch size of $100$ for all of the datasets. For gray-scale datasets We dynamically binarize each training data, but we do not binarize the exemplars that serve as the prior. We utilize early stopping for training VAEs, where we stopped the training if for $50$ consecutive epochs the validation ELBO does not improve. We use $40$ dimensional latent spaces for gray-scale datasets while using $128$ dimensional latent for CelebA. To limit the computation costs of convolutional architectures, we considered kNN based on euclidean distance in the latent space, where $k$ set to $10$ for gray-scale datasets and $5$ for CelebA. The number of exemplars set to the half of the training data except in the ablation study section.

## M  Misclassified MNIST Digits

A classifier trained using exemplar augmentation reached average error of $0.69\%$. Here we show the test examples misclassified.

Figure 5: Misclassified images from MNIST test set for a two layer MLP trained with Exemplar VAE augmentation.

## Footnotes

[1]https://github.com/jmtomczak/vae_vampprior