[Reviews · NeurIPS 2020]

Review 1

Summary and Contributions: The paper proposes a new class of VAEs that introduces nonparametric notion by including training data. Then, the idea boils down to a specific choice of the marginal over latents that is a mixture distribution (the number of components = the number of training data). In order to overcome overfitting, the authors propose two regularizers. Moreover, in order to allow efficient training, the authors utilize kNN procedure to choose a subset of training data. The experiment clearly show an advantage of the proposed approach over the standard marginal over z's and the VampPrior.

Strengths: + The idea of adding a nonparametric flavor to VAEs is interesting. + The paper is clearly written. + The presented concepts are explained in a lucid manner.

Weaknesses: - The presented idea is closely related to: Graves, A., Menick, J., & Oord, A. V. D. (2018). Associative compression networks for representation learning. arXiv preprint arXiv:1804.02476. It would be beneficial to comment on similarities and dissimilarities between this paper and the presented approach. - Since the approach uses training data, it would be insightful to provide training/inference wall-clock time for a vanilla VAE (w/ Gaussian prior and w/ VampPrior) and the proposed VAE, and also provide a simple computational complexity analysis. This would greatly help to see what is the tradeoff between better bpd/quality of images and higher complexity.

Correctness: The presented method seems to be correct, I cannot find any flaw or unclear choice. All steps are well motivated.

Clarity: Yes, the paper is clearly written. All concepts are easy to follow.

Relation to Prior Work: The prior work is clearly presented and proper paper are cited. However, one paper is definitely missing: Graves, A., Menick, J., & Oord, A. V. D. (2018). Associative compression networks for representation learning. arXiv preprint arXiv:1804.02476. I would expect a proper discussion of this paper.

Reproducibility: Yes

Additional Feedback: o The idea of leave-one-out during training seems to be equivalent to the pseudolikelihood approach. o Please correct "Algorithm ??" in the line 140. ===AFTER THE REBUTTAL=== I would like to thank the authors for their rebuttal. After the discussion with the other reviewers, I decided to keep my score. The paper is well-written, and the idea is interesting, however, as indicated in the reviews, the novelty is somehow limited. And since the ideas are rather incremental, a more convincing experiments are necessary. o The authors present samples from CelebA, but no bpd is reported. I would suggest (at least in the appendix) to indicate what bpds are for the proposed VAE, VAE w/ Gaussian, VAE w/ VampPrior. I do not expect SOTA scores, but I would appreciate these scores on RGB images, not only black&white or gray-scale images.


Review 2

Summary and Contributions: The paper introduces Exemplar Variational Autoencoders, an extension of VAE that utilizes a non-parametric approach for training and sampling. The model re-uses an encoder network to embed exemplar data points and Parzen window estimator with Gaussian kernel for a latent prior. To reduce computational complexity arising from a large number of exemplars needed to estimate the value of exemplar prior, the authors use an approximate k nearest neighbour search for the most influential points in latent space justifying this as a lower bound on the prior. To avoid overfitting (a simple reconstruction of exemplars), the authors propose two regularization techniques, namely leave-one-one training and exemplar subsampling, both resulting in lower-bounds of the original unregularized KL divergence between the variational posterior and the exemplar prior (essentially a Gaussian mixture on exemplar points). Density estimation experiments show marginal improvement, although other applications (e.g. data augmentation for classification) seem promising.

Strengths: The paper has an extensive experimental section showing benefits in a number of applications. For example, data augmentation with Exemplar VAE is shown to reduce error rates on the classification task, which is not as prominent in the case of VAEs with Gaussian prior and VampPrior. Apart from that, I believe that bridging parametric and non-parametric approaches is a relevant direction to explore.

Weaknesses: I might be wrong, but I tend to see the introduced exemplar-based prior as a way of constraining model towards training data and reducing generalization. The authors did a great job introducing regularization techniques, but might it be that these techniques would also boost original VAE and VAE with VampPrior? It is also not discussed how one should choose k and M hyperparameters in kNN search and exemplar subsampling.

Correctness: There are few things to address in experiments. 1. The reported scores for VampPrior (Table 3) are different from the original paper (Tables 1,3,4) although it seems that the experimental setting is the same (40d latent space, architectures, etc.). For example, the authors of VampPrior report NLL = -101.18 for HVAE on OMNIGLOT while the authors of this paper report -103.30. Looking at the original VampPrior paper, the improvement of ExemplarVAE in density estimation disappears. 2. The authors report an FID score = 39 on 64x64 CelebA, saying that this result is possible with a post-processing step of reducing the variance in the latent prior. However, it is not compared to VAE with VampPrior with or without similar post-processing. Also, wouldn't this post-processing step limit the diversity of samples? How can it be justified? 3. When testing different ratios of M/N in exemplar subsampling, it doesn't seem like changing M has an apparent effect on the model. 4. I believe that Exemplar VAE data augmentation is complementary to the methods that are reported in Tables 5,6, so it should not be a problem to add two outperforming methods that are mentioned in the text. Also, what is the performance of Exemplar VAE without Label Smoothing?

Clarity: The paper has many typos, I recommend proofreading it for these minor errors. It would also be better to make references to specific sections in an appendix rather than generally referring to supplementary materials. It would also help including pseudo-code of the method to help understand how Exemplar VAE works.

Relation to Prior Work: Yes, I believe it is well discussed.

Reproducibility: Yes

Additional Feedback:


Review 3

Summary and Contributions: The paper proposes a variant of VAE with a Parzen window prior in the latent space. Data augmentation can be done with this VAE, using latent space nearest neighbor relationship, for boosting image classification performance. Using a non-parameteric Parzen window is also found to perform better than traditional factorized Gaussians. Practical tricks for reducing computational requirements in Exemplar VAEs are also presented.

Strengths: 1. The intuition is based on generating prior distribution based on statistics of the training samples. The RAT training algorithm is in line with the intuition. 2. The authors use random sampling and kNN search to reduce computational consumption in estimating the prior distribution. 3. The proposed method works across a range of problems: density estimation, representation learning, and data augmentation. 4. Experiments are set up decently.

Weaknesses: 1. In line 117-118, the authors claim that "an exemplar prior with a shared isotropic covariance does not greatly impact the expressive power of the model." However, no experimental results are available. 2. In ablation study (Section 5.1), the authors only show the impact of sampling size M, what about the effect of k in kNN? 3. There is no clear description of the generation process. (How to sample from the parzen window prior?) 4. There should be more comparison with other variants of VAE. 5. In RAT algorithm, should it be "kNN = Cache.kNN(z) \cap π"? What if Cache.kNN(z) and π have no common elements? 6. In the data augmentation experiments, why only augment xi with the expectation of r(z | xi)? What if we draw latent code from r(z | xi)? 7. The examplar subsampling and kNN are proposed to deal with large training datasets. There lacks experimental results with large datasets like imagenet to validate this algorithm.

Correctness: The experiments are well tested on the given datasets. However, most empirical results are only demonstrated on small datasets, though the paper claims that it can scale to large datasets.

Clarity: The paper is well presented and organized around the exemplar VAE. However, a lot of details are in supplemental materials, for example the RAT algorithms.

Relation to Prior Work: The proposed mehtod is a variant of VAE, but modified in aspects like the prior and the sampling method, in the spirit of combining the exemplar-based approaches with distribution-regularized approaches.

Reproducibility: Yes

Additional Feedback:


Review 4

Summary and Contributions: This paper proposes to replace the Gaussian prior in a standard VAE with a uniform mixture of Parzen window prior centered on each training data point (exemplar). Heuristics based on a leave-one-out objective and exemplar sampling (a random subset of the training set) is used to combat overfitting.

Strengths: Unlike VampVAE with learned exemplars, the exemplars in the proposed method are sampled from the training data and the prior calculation for each z can be accelerated by approximate nearest neighbor search. Experiments demonstrate that the proposed method Exemplar VAE outperforms VAEs with a Gaussian prior and slightly outperforms VampVAE Experiments on MNIST and Fashion-MNIST show that Exemplar VAE is effective in generating augmented data to learn good representations for downstream tasks.

Weaknesses: My main concern is about the novelty and significance of this work. The proposed method Exemplar VAE is a trivial variant of VampVAE: In VampVAE, exemplars are learned; in Exemplar VAE, the exemplars are randomly sampled from the training set. Although this work is highly similar to VampVAE, the presentation of this paper is not as clear as VampVAE. As already shown in VampVAE, it is not surprising that VampVAE (Exemplar VAE) beats VAE with a standard Gaussian prior. A better baseline is VAE with a Gaussian mixture prior that has already been thoroughly studied for clustering and data visualization in the literature. Compared to the further hierarchical prior development in the VampVAE paper conducted several years ago, the simple modification presented in this work is not significant enough.

Correctness: The method is technically correct.

Clarity: The paper is well written in general.

Relation to Prior Work: Yes.

Reproducibility: Yes

Additional Feedback: I read all the reviews and the rebuttal. I agree with the authors that the proposed method is different from learned pseudo-exemplars in the embedding space as in VampVAE, and this work uses real exemplars in the image space. However, I am not convinced that randomly sampling exemplars in the data space with some heuristics based on LOO and trivial exemplar subsampling as regularizations on toy datasets is a significant contribution extending the exemplar-based prior in VampVAE. A possible limitation of the proposed Exemplar VAE is that, the generative model might not learn much beyond reconstruction, instead, it only produces some random samples that stay close to epsilon-ball of training data points. It's possible that Exemplar VAE even performs no better than a deterministic autoencoder with tiny Gaussian noise added to latent codes and k-means regularization in the latent space. VampVAE doesn't have this issue. Moreover, demonstrating the representation learning capabilities and data augmentation benefits (an orthogonal contribution) of Exemplar VAE on (F-)MNIST is not interesting. If this work wants to demonstrate the significance of Exemplar VAE, large-scale experiments on more challenging datasets such as ImageNet that do require approximate NN search should be conducted. In sum, considering previous works on exemplar learning as a prior, marginal improvement over VampVAE, and small-scale non-challenging experiments that doesn't require approximate nearest-neighbor search, I am not convinced that the work in its current form is a significant contribution to NeurIPS.

[Author Response · NeurIPS 2020]

1 **Author Response for "Exemplar VAEs for Exemplar based Generation and Data Augmentation"**

2 We thank the reviewers for their valuable feedback. We respond to the main comments below, starting with the most critical review.

3 **R4**: *"My main concern is about the novelty and significance of this work..."* The reviewer's assertion about limited novelty may stem from a characterization of novelty in terms of modeling modifications. We claim no credit for the idea of using a mixture of variational posteriors as the prior, since it is known in the varitional inference community that the Bayes optimal prior for a VAE is a mixture of variational posteriors *a.k.a.,* aggregated posterior [2]. That said, previous work [3, 1] makes the observation that the use of the aggregated posterior as a VAE prior yields poor performance and massive overfitting, hence remedies such as learning a limited number of pseudo inputs were proposed [3]. This paper shows that the aggregated posterior is indeed an excellent prior for VAEs when simple yet **novel regularizers** are used. We propose a **new log marginal likelihood lower bound** based on kNN retrieval to enable the use of a large number of mixture components in the VAE's prior. Importantly, we propose a **novel application of VAEs to data augmentation** to improve supervised learning and show that the nonparametric nature of the Exemplar VAE is effective for improving classification error. Demonstrating the effectiveness of generative data augmentation is important as no prior work on VAEs has shown similar gains. These contributions are significant for the generative modeling community as they bridge the gap between likelihood based generative models, nonparameteric exemplar based approaches, and data augmentation strategies.

15 **R4**: *"...not surprising that VampVAE (Exemplar VAE) beats VAE with a standard Gaussian prior. A better baseline is VAE with a Gaussian mixture prior...studied for clustering and data visualization..."* Note that VAE with a VampPrior [3] outperforms VAE with a Guassian mixture prior, so we compared to VampPrior. But we'll include a comparison with MoG prior in the final revision.

18 **R2**: *"...I tend to see the introduced exemplar-based prior as a way of constraining model towards training data and reducing generalization"* Our empirical results suggest that Exemplar VAEs outperform predominant VAE variants in terms of generalization to held-out test sets as measured by log-likelihood. An intuitive explanation of this result is that learning to augment existing examples into new ones is easier than learning to generate examples from scratch.

22 **R2**: *"did a great job introducing regularization techniques, but might it be that these techniques would also boost original VAE and VAE with VampPrior?"* Good question. Our most important regularizer, leave-one-out, does not apply to parameteric priors such as VamPrior. Our preliminary experiments suggest that exemplar subsampling in a VampPrior has a similar effect to reducing the number of pseudo-inputs, but we will include an ablation and a discussion in the paper to address this issue.

26 **R2, R3**: *"...not discussed how one should choose k and M hyperparameters"* Section 5.1 (line 223) includes an ablation study to analyze the impact of $M$ proportional to the dataset size $N$. We conclude that $M = N/2$ is a reasonable choice. We select $k$ based on our computational budget to match the training cost of related work. Ignoring computation cost, a larger value of $k$ is preferred.

29 **R2**: *Misc.* Thank you for such detailed comments. 1) To replicate VampPrior's results, we used the publicly available official repository, which gives consistent numbers on dynamic MNIST, but results in some discrepancy on Omniglot. The use of a small validation set (2K images) for early stopping can explain the discrepancy on Omniglot, and it is possible that VampPrior used a different procedure for early stopping on Omniglot. Nevertheless, our comparison is fair and all techniques use the same early stopping, training, and validation procedures. Importantly, note that for the ConvHVAE architecture, which has the best likelihood numbers, our replicated VampPrior numbers are better than the original paper. 2) As observed by prior work [4], VampPrior on CelebA didn't converge to a good solution in our experiments, which is the reason we didn't report VampPrior's numbers. The problem may be due to the initialization of pseudo inputs, which is a limitation of VampPrior. It's common to decrease the temperature of the model to improve sample quality and FID scores. 3) The choice of $M < N$ results in a consistent improvement on MNIST and Omniglot, so exemplar subsampling is helpful. 4) Agreed that Exemplar VAE augmentation can be combined with other approaches to reach a better classification error. MLPs are not competitive on permutation invariant MNIST without label smoothing. We'll fix the typos and include references to sections of the appendix. Part 9 of the appendix presents the pseudo-code.

41 **R1**: Thank you for bringing Graves *et al.* (2018) to our attention. We'll discuss in the final revision. Graves *et al.* learn an ordering of the data points focusing on autoregressive decoders to decrease the description length of transmitted codes. They propose a conditional prior that is difficult to compare against without an ordering. They also propose an unconditional prior that does not yield any likelihood gains. By contrast, we define a generic prior, which can be used to define an ordering to achieve the goal of Graves *et al.* and provides likelihood gains. Our unsupervised classification score outperforms Graves *et al.* on MNIST (98.5 *vs.* 98.87), which suggests our representations are higher quality. Finally, the summary of our contributions above is orthogonal to Graves *et al.*

47 **R1**: *"wall-clock time"* The cost of training Exemplar VAE is similar to VampPrior when the number of exemplars per minibatch is equal to the number of pseudo inputs, *e.g.,* for ConvHVAE on Omniglot with a minibatch size of 100 on a single GPU, VampPrior with 1000 pseudo inputs takes *58s/epoch* and Exemplar VAE with 10-NNs takes *51s/epoch*. ConvHVAE on MNIST & FashionMNIST with VampPrior takes *82s/epoch vs. 107s/epoch* for Exemplar VAE, since VampPrior uses 500 pseudo inputs here.

51 **R1**: *present samples from CelebA, but no bpd is reported...* ELBO numbers for CelebA are reported in the appendix, part 2. We can transform these numbers to bpd if that's more desirable. Also, there is some similarity between pseudo-likelihood and leave-one-out in Exemplar VAE, but pseudo-likelihood is one dimension given the rest, whereas leave-one-out is one example given the rest.

54 **R3**: *Misc.* The generation process is explained at the beginning of Section 3. For exemplar data augmentation we indeed sample from $r(z \mid x_i)$. CelebA has close to 200k 64x64 images, so we validated the effectiveness of method on a decent scale dataset.

# References

[1] Bornschein, Mnih, Zoran, and Rezende. Variational memory addressing in generative models. *NIPS*, 2017.

[2] Hoffman and Johnson. ELBO surgery: yet another way to carve up the variational evidence lower bound. *NIPS Workshop*, 2016.

[3] Tomczak and Welling. VAE with a VampPrior. *AISTATS*, 2018.

[4] Xu, Luo, Henao, Shah, and Carin. Learning autoencoders with relational regularization. *arXiv:2002.02913*, 2020.


[Meta-Review · NeurIPS 2020]

*PROS: add a nonparametric flavor to VAEs, which is interesting. Extensive experimental section showing benefits in a number of applications. *CONS: the novelty is somehow limited. And since the ideas are rather incremental Meta-reviewer recommendations: The paper is borderline. R2 is considering increasing the score. R4 recommends rejection based on the lack of novelty compared to the VampPrior and that the paper conducts small-scale non-challenging experiments that doesn't require approximate nearest-neighbor search. He proposes to run the method on ImageNet but I believe this cannot be a condition for acceptance since not everyone has the potential for running such experiments. Furthermore, the paper already covers quite a lot in experiments with MNSIT, Omniglot, FashionMNIST and CelebA, and classification. I believe that R4's concern for novelty are successfully addressed in the rebuttal. The paper shows that the aggregated posterior is an excellent prior for VAEs when simple yet novel regularizers are used. They propose a new log marginal likelihood lower bound based on kNN retrieval to enable the use of a large number of mixture components in the VAE’s prior. They also propose a novel application of VAEs to data augmentation.